



# Attribution of the role of climate change in the forest fires in Sweden 2018

Folmer Krikken[1], Flavio Lehner[2], Karsten Haustein[3], Igor Drobyshev[4,5,6], Geert Jan van Oldenborgh[1]

[1]Royal Netherlands Meteorological Institute (KNMI), De Bilt, Netherlands,
[2]National Center for Atmospheric Research, Boulder, USA
[3]Environmental Change Institute, University of Oxford, Oxford, United Kingdom
[4]Centre for Forest Research, Montreal, Canada
[5]NSERC-UQAT-UQAM Industrial Chair in Sustainable Forest Management Universite du Quebec, Canada
[6]Swedish University of Agricultural Sciences, Southern Swedish Forest Research Centre, Alnarp, Sweden

*Correspondence to*: Folmer Krikken (folmer.krikken@knmi.nl)

**Abstract**. In this study we analyse the role of climate change in the forest fires that raged through large parts of Sweden in the summer of 2018 from a meteorological perspective. This is done by studying the Canadian Fire Weather Index (FWI) based on sub-daily data, both in reanalysis datasets (ERA-Interim, ERA5, JMA55 and MERRA2) and three large ensemble climate models (EC-Earth, W@H and CESM) simulations. The FWI based on reanalysis correlates well with observed area burned in summer (r=0.6 to 0.8). We find that the maximum forest fire risk in July 2018 had return times of ~24 years for Southern and Northern Sweden. Further, we find a negative trend of the FWI for Southern Sweden over the 1979 to 2017 time period, yielding a decreasing risk of such an event solely based on reanalysis data. However, given the short observational record, large uncertainty between the reanalysis products and large natural variability of the FWI we cannot draw robust conclusions from reanalysis data.

The 3 large-ensembles with climate models on the other hand point to a roughly 1.1 times increased risk for such events in the current climate relative to pre-industrial climate. For a future climate (2C warming) we find a roughly 2 times increased risk for such events relative to pre-industrial climate. The increased fire weather risk is mainly attributed to the increase in temperature. The other main factor, precipitation during summer months, is projected to increase for Northern Sweden, and decrease for Southern Sweden. We however do not find a clear change of prolonged dry periods in summer months that could explain the increased fire weather risk.

In summary, we find a small but positive role of global warming up to now in the 2018 forest fires in Sweden, but a more robust increase in the risk for such events in the future.



## 1 Introduction

The summer of 2018 in Sweden was characterized by numerous large forest fires spread over large parts of the country.
Though forest fires are common in Sweden, the number of fires and total area burned in 2018 were much higher than observed over recent years (2008-2017; Figure 1). Spring and summer weather conditions in 2018 were anomalously dry and warm. This was caused by very persistent atmospheric blocking, especially in May and July. In July the high surface pressure (Figure 2a) caused high temperature (Figure 2b) and anomalously little precipitation (Figure 2c) over northwestern Europe. The high temperature and lack of precipitation resulted in high forest fire risk over the whole of Scandinavia.
Especially in Sweden, this gave rise to numerous forest fires with a total burned area of more than 20.000 ha.

An often-raised question during and after such extreme events concerns the possible influence of climate change, i.e., has climate change made such an event more or less likely? Hence, climate attribution studies of extreme weather events is a rapidly increasing field of research, with analysis on e.g. extreme precipitation events (Oldenborgh et al., 2017), heat waves (
Sippel et al., 2016), droughts (Hauser et al., 2017) and storms (Vautard et al., 2019), where in many cases there was indeed evidence of increased risk of extreme weather due to climate change (Schiermeier, 2018). For forest fires, the first attribution of climate change on forest fires in Canada was already found by Gillett et al. (2004) based on the CGCM2 model. Abatzoglou and Williams (2016) found that for the western United States human-caused climate change more than halved humidity of forest fuels since the 1970s and doubled the cumulative area of forest fires since 1984. A recent study by
Kirchmeier-Young et al., (2018) found a strong influence of climate change on the 2017 British Columbia wildfires, with such events being 2-4 times more likely with climate change in the CanESM2 model. Abatzoglou et al. (2018) found that increases in extreme fire weather days due to anthropogenic climate change are evident on 22% of burnable land area globally.

For Sweden specifically, Yang et al. (2015) found that in a future climate there is an increased risk of forest fires in Southern Sweden but a decreased risk in Northern Sweden using the downscaled and bias corrected ECHAM5 climate model. Also, for the neighbouring country of Finland, climate model projections point to an increased risk of forest fires (Lehtonen et al., 2016).

Here we analyse the connection between the 2018 extreme forest fire season and climate change using large ensembles of multiple climate models. As characteristics of regional precipitation and droughts can be highly model dependent (Hauser et al., 2017), it is crucial to use multiple climate models for this analysis. To the best of our knowledge, this is the first such multi-model framework applied to an attribution study of an extreme forest fire event.



Note that we only analyse the meteorological aspect of this event, and not other aspects relevant for such extreme events such as the sources of ignition and the influence of fire mitigation strategies. Hence, in our analysis we take forest fire risk to be equivalent to fire weather risk. We do investigate to what extent the fire risk corresponds to actual area burnt.

## 2 Data and Methods

### 2.1 Fire weather risk

The metric used to quantify forest fire risk is the Canadian Fire Weather Index (FWI; Van Wagner, 1987). This is a weather-based system that models soil moisture at 3 different depths, and, based on the upper soil moisture content and wind speed, creates an estimate for the initial spread rate of fire. It is based on four meteorological variables, namely local noon temperature (T2M), relative humidity (RH), surface wind speed and 24 hour cumulative precipitation. Though this metric

was developed and tuned for the Canadian boreal region, it also performs well over Sweden (Gardelin, 1997; Yang et al., 2015).

### 2.2 Statistical methods

In event attribution studies, the first step is to define the event in such a way that is best reflects the impact of the event. In

section 3 we will discuss how we define this event in more detail. The second step, in order to assess the rarity of this event, is to fit a Generalized Extreme Value Distribution (GEV) function on a sample of block (yearly) maxima extracted from a FWI time series. The GEV function is described by three parameters: the position parameter $\mu$, the scale parameter $\sigma$ and the shape parameter $\xi$.

In order to assess the risks of certain events during previous climate based on the trend in observations and reanalysis products, we fit the observed FWI to a GEV that depends on the smoothed (4th order polynomial) global mean surface temperature (GMST). Here, GMST is taken from the National Aeronautics and Space Administration (NASA) Goddard Institute for Space Science (GISS) surface temperature analysis (GISTEMP, (Hansen et al., 2010). This results in a distribution that varies continuously with GMST. This distribution can be evaluated for a GMST in the past (e.g., 1950 or

1900) and for the current GMST. A 1000-member non-parametric bootstrap procedure is used to estimate confidence intervals for the fit.

For the FWI we choose the dependence of the FWI the same way as for precipitation (described fully in Wiel et al., 2017): the position and scale parameters $(\mu,\sigma)$ have the same dependence on GMST so that their ratio (also called the dispersion



parameter) is constant. The dependence is exponential for precipitation, $\mu(T) = \mu_0 \exp(\alpha T/\mu_0)$, $\sigma(T) = \sigma_0 \exp(\alpha T/\mu_0)$, we use the same here.

We use the risk ratio to quantify the impact of climate change on the FWI. This ratio, calculated as the risk of an event occurring during the current or future climate divided by the risk of an event occurring during pre-industrial climate

conditions, indicates how much more or less likely a certain event will be relative to pre-industrial climate. Thus, a risk ratio of 2 means an event will be 2 times more likely relative to pre-industrial climate.

## 2.3 Reanalysis

We use multiple reanalysis dataset as an estimate of the observed state, namely ERA-Interim (ERA-I, Dee et al., 2011),

ERA5 (Copernicus Climate Change Service, 2017), the Japanese 55-year Reanalysis (JRA-55, Kobayashi et al., 2015) and the Modern-Era retrospective analysis for Research and Application, Version 2 (MERRA2, Global Modeling and Assimilation Office, 2015). The advantage of using reanalysis datasets compared to in-situ observations is that it provides an observationally constrained continuous gridded dataset, enabling direct comparison to climate model output. The orography of Sweden is such that the relatively low-resolution models used to generate the reanalysis can represent the weather well in

this area. We use multiple reanalysis products in order to sample the uncertainty in these products. All products provide a continuous dataset from 1979/1980 to current, with the exception of JRA-55 which spans the period from 1955 to current.

## 2.4 Models

We use climate model simulations from three different coupled climate models with large ensembles: EC-Earth v2.3

(Hazeleger et al., 2010; 2011), the Community Earth System Model version 1 (CESM1, (Kay et al., 2014) and Weather@Home (W@H, Guillod et al., 2017; Massey et al., 2015) (Table 1). The EC-Earth and CESM are large ensembles of transient climate simulations with historical forcing prior to 2006 and the RCP85 forcing (Riahi et al., 2011) from 2006 onwards. The W@H climate simulations are two different simulation, one with the actual observed forcing to represent current climate, and one with natural forcing only (i.e. no anthropogenic forcing) to represent pre-industrial climate. From

EC-EARTH and CESM  we select three periods that (1) describe the unperturbed climate (i.e., pre-industrial climate), (2) the current climate and (3) the 2°C warming threshold (future climate).The pre-industrial, current and future climate states are hereafter referred to as respectively PI, 1C and 2C. Note that 'current climate', or '1C', is chosen in a way to best compare to reanalysis (which covers the years 1979-2018) as described below.

First, we select the time periods from EC-Earth and CESM1 that represent the same incremental global warming from 'PI' to '1C' as in observations: in GISTEMP, GMST increases by 0.67°C between 1900-1950 and 1979-2018. Finding the same





warming increment in EC-EARTH and CESM results in the time periods listed in Table 1 for the two models. For the '2C' climate we select a 30 year window with a 2°C warming relative to PI (Table 1). For the W@H simulations the GMST increase between the 'natural forcing' simulations and the 'actual forcing' simulations is 0.65$^0$C, which is very close to the
observed warming.

A second bias correction step is performed on the basis of return times of the specific event and can be seen as a local bias correction in contrast to the first step, which aimed at aligning simulations and observations with regard to the level of global warming. We first calculate the return time of the event from observations or reanalysis (Figure 3). In the '1C' model
simulations we then select the FWI that corresponds to that specific return time. This FWI value is used to estimate return times in the other simulated climate states (PI and 2C). The advantage of this approach is that it preserves the spatiotemporal consistency of the simulated fields, the relation among the meteorological variables, and it makes no assumptions on non-stationarity in bias correction, which are typical issues in (multi-variate) bias correction methods (Ho et al. 2011; Ehret et al. 2012). Note that this bias correction is only a viable method if the risk ratio is not too sensitive to the event return time,
which is the case here (not shown).

We calculate the FWI on the original grid of the models. Since Southern Sweden has a different fire climate than Northern Sweden (Drobyshev et al., 2012), we calculate spatial averages for northern (Norrland), middle (Svealand) and southern (Götaland) Sweden (Figure 4a). Since high fire weather risk events are mainly associated with large high pressure systems, it
is important to validate the persistence of high pressure systems in the climate models by comparing it to reanalysis data. Following the method of Pfleiderer and Coumou (2018), which represents persistence as the number of consecutive warm days, we find that the models are in good agreement with reanalysis in respect to persistence of high pressure systems (not shown).

For EC-Earth we compute the FWI based on local noon data (12 UTC), but for the CESM-LE and W@H sub-daily data is not available. Hence, for these models we compute the FWI based on daily average wind speed and humidity, daily maximum temperature and daily cumulative precipitation. While this approach is common (Abatzoglou et al., 2019), results can differ between both methods especially for fire danger extremes (Herrera et al., 2013). In order to assess whether this has an effect on our analysis we tested the influence of using local noon data, or daily average combined with maximum
temperature for EC-Earth. Though the values of the FWI do differ, there are no significant differences for both methods on the calculated risk ratios. Hence, we assume that using daily maximum temperature and daily average values for the other variables for the calculation of the FWI in CESM-LENS and W@H does not affect the calculated risk ratios significantly.





## 3 Event definition

We first investigate whether the FWI is a good proxy for actual fires in Sweden. For the event definition we use ERA-
Interim as the observational estimate. The FWI is a physical approximation of climatological fire risk and it has been found
to be a robust proxy for actual fires (Wotton, 2009). However, there can be a strong seasonal dependence on the correlation
between the FWI and actual fires (Lehtonen et al., 2016). We test this for Sweden by studying the correlation between the
FWI and observed area burned (MSB, 2016) for the period 1996 to 2012 (Figure 4b). Note that, here we leave out Svealand
because there are insufficient fires to compute the relevant statistics. The results show that there is indeed a strong seasonal
dependence on the relation between monthly averaged FWI and area burned, with generally high correlations from July
onwards, but lower correlations for April to June for Norrland and May for Götaland. These findings correspond to the
findings from Lehtonen et al. (2016), who relate the low correlation in spring to the influence of more human-caused fires,
whilst in summer natural ignitions is a more important ignition source thus yielding a stronger relation with weather
variables.


Next, we analyse the FWI for all three regions, using ERA-Interim (Figure 5) and the observed area burned (Figure 1). There
were two distinct periods of high FWI (above the 95% quantile), namely in late May to early June and in July. Interestingly,
the values in May were even higher than those observed in July, although the actual area burned was much higher in July.
This indicates a possible pre-conditioning (drying out the soil) of summer FWI by the occurrence of a dry spring. Note that
the pre-conditioning by the dry spring is still included in this event definition because the FWI calculation includes an
estimate of the moisture content in the deeper soil layer. This moisture content, estimated by the Drought Code within the
FWI calculation, includes memory of ~52 days (Van Wagner, 1987).

Based on the findings from Figure 1, 4 and 5, we define our event as the maximum 7-day running mean FWI in the months
of July and August, disregarding the FWI peak in May due to much lower correlations with area burned, probably due to
lower ignition rates. The 7-day running mean is applied as fires are more likely to happen during prolonged period of high
fire weather risk, whilst still holding enough independent samples per year for a robust GEV fit. Though June also shows
relative high correlation with observed area burned, the strong fires were mainly in the summer months.

## 4. Results

### 4.1 Reanalysis

As previously stated, July 2018 was characterized by a large persistent high pressure area over Northern Europe (Figure 2),
yielding high temperatures, little precipitation and moderate winds. The meteorological conditions for fire weather were thus
quite extreme. This is quantified by the high return times of such conditions for July 2018 (Figure 6). These values are based



on a GEV-fit, based on the maximum value of FWI in July and August for every year, with a 7-day running mean applied.
This fit assumes that the climate does not change over time.

It is striking to see that, although all reanalysis products are constrained by observations, there are still quite large differences in the FWI value for the 2018 event and the associated return times. For Norrland, we find large significant differences between JMA-55, with a return time of ~5 yr, and ERA-I, ERA5 and MERRA-2 with return times of ~30 yr. Also for
Svealand there are rather large differences in return times, with ~10 years for JMA-55 and ~20 yr for ERA-I and ERA5 and ~50 yr for MERRA-2. In Götaland we also find differences between the products, but now JMA-55 closely matches ERA5 with return times of ~8-10 yr, and ERA-I has a higher return times of ~20 yr and MERRA-2 even higher at ~60 yr. Note that the uncertainties on these return times (denoted by the horizontal bars) are large but almost completely correlated across datasets as they derive from the same natural variability (except JRA-55 that includes more data). An analysis on the
meteorological variables used in the FWI reveals that it is mainly precipitation that causes the differences in FWI and return times across products. For MERRA2, it is also related to a generally lower temperature (not shown).

These results stress the importance of using multiple reanalysis products in order to get a better estimate of the observed event and its associated uncertainty. For bias correction of the climate model data, we use the average of all four return times
from the different reanalysis products.

In order to analyse whether such an event has become more or less likely relative to a climate without anthropogenic emissions ("pre-industrial", PI), we fit the yearly maximum FWI to a GEV that scales with the smoothed GMST (as described in the methods section). We can then evaluate the probability of such an event conditional on different climate
states as defined by GMST. Figure 7a shows the risk ratios for the reanalysis products. Note that, as stated before, the risk ratio tells us how much more or less likely such an event has become today ('1C' climate) relative to PI climate. The reanalysis data shows a slightly decreased risk of high FWI events for all three regions for the 1C climate. This is due to a negative trend of July and August FWI over recent decades for these regions. However, the very large uncertainties easily encompass one (no change), indicating that this is most likely a spurious trend caused by natural variability. This trend is
largely absent in JMA-55 which has a 25 year longer time span than the other reanalysis datasets. Hence, we cannot draw any robust conclusions from the trends of the reanalysis dataset alone.

## 4.2 Models

With large ensemble climate model data we can circumvent the problems of undersampled natural variability, allowing us to
get more robust estimates of whether the likelihood of such an event changes with time. Figure 7b shows the risk ratios of


the climate models for present climate (1C) relative to PI climate and future climate (2C) relative to PI climate. First we will focus on the comparison of 1C to PI.

The model W@H shows a small, but significant increased risk of approximately a factor 1.5 for such events for all three regions. EC-Earth shows no clear change in risk for such an event, with risk ratios close to 1, whereas CESM does show a
small increase in risk, though not significant. On average, we find a small (not significant) increase in risk for all three regions. In the 2C climate, the risk ratios increase more strongly relative to PI climate. CESM shows significant increased risk ratios of ~3, with the largest increase in risk in Norrland. EC-Earth also shows an increased risk, though not as large as CESM. On average for all three regions we find a risk ratio of ~2

**5 Discussion**

In general, we find a factor 1.2 increased risk for such events for current climate relative to PI climate, and a significant increase in risk of factor 2 to 3 for a 2°C warmer climate relative to PI climate. To better understand why there is an increased risk of such events, we investigate the individual meteorological variables at the time of the maximum July and August FWI in models (Figure 8).


All models show a clear trend towards higher temperatures, which is unsurprising as present day and future climate are chosen as ~1°C and 2°C warmer climates. The increase in temperature between 1C and 2C is generally much larger than 1°C, especially in CESM-LE under 2C, because land heats up faster than the global mean. This can partly explain the relative strong increase in fire risk in CESM-LE for future climate. In EC-Earth the relative humidity seems to reflect the changes in
precipitation where it increases from PI to 1C and then decreases slightly in the 2C climate. In CESM-LE, we find no clear change in 1C, but a decrease of RH in 2C. W@H also show a small decrease in RH relative to PI climate. For the wind speed, the differences between the climate states are very small and do not affect changes in the FWI appreciably. We further subset the model values by focusing on FWI events larger than the 2018 observed event (circles in Fig. 8) to investigate whether certain variables are predominantly affecting extreme FWI. We find no relationship between wind and extreme FWI
values , indicating wind is not an important explanatory variable for extreme FWI events over Sweden.

For precipitation, we generally find an increase from PI to 1C for all regions and climate models, with the exception of W@H with a small decrease in precipitation. For 2C, however, there are strong differences between the regions, where in Norrland precipitation further increases whilst for Svealand and Götaland it decreases towards PI values (EC-Earth) or stays
constant with present climate values (CESM-LE). Note however, that the precipitation values associated with high FWI values (circles in Figure 8) do not show this upward trend for Norrland. Hence, changes in mean precipitation do not necessarily reflect the changes in prolonged dry periods. An analysis of the trends in the lower (dry periods) and middle quantiles of 30-day precipitation in summer shows clear changes in the median but no clear changes in the lower quantiles


(not shown). This is also demonstrated by Pendergrass et al. ( 2017), who find that precipitation variability generally
increases in a warmer climate.

The changes in FWI between the different climates relate mostly to changes in precipitation and temperature, as RH follows
the changes of these variables. The higher temperatures for present and future climate relative to PI climate yield an increase
of FWI. However, the increase of precipitation can counteract this increase. The fact that FWI increases even in models with
mean increases in precipitation shows that temperature increases dominate future increases in FWI.

One potential reason for the non-linear model behaviour with regard to the different warming thresholds is the non-linear
evolution of the radiative forcing. Global dimming associated with the release of cooling anthropogenic aerosol particles
(Wild 2009) has effectively offset the greenhouse gas induced warming during 1950-1975. In contrast, global brightening
has accelerated the warming thereafter, particularly over North America and Europe where sulphate aerosols emissions were
curbed substantially. As a result, aerosols may have suppressed rainfall during the dimming phase, but invigorated
precipitation during the subsequent warming until now, exceeding the direct temperature effect. The smaller risk reduction in
the longer JMA-55 time series would support this hypothesis.

Another dynamic factor that is projected to come into play over the next decades is the development of a heat low over the
Mediterranean area. This would increase the possibility of easterly wind over northern Europe and hence dry weather,
offsetting the trend towards wetter summer weather up to now (Haarsma et al., 2009).

Our results mostly agree with previous research. The work of Flannigan et al. (2012) points to increased risk of forest fires
over the whole of Sweden for multiple climate projections. Findings from Yang et al. (2015) point to an increased risk of
forest fires in the southern part of Sweden, but not for the northern part where they point to increased precipitation which
reduces fire risk. This difference can be caused by undersampling of extreme events, since Yang et al use a single 30 year
time slice of future (2071-2100) climate. It must also be noted that local future precipitation trends are highly uncertain
(Lehtonen et al., 2016), implying that using only one climate model for future projection (as in Yang et al., 2016). Even the
three climate models in this study will likely underestimate the model uncertainty in the precipitation trends. Hence, future
work should focus on using more large ensemble climate models in order to better sample the uncertainty in the future
climate projections.

Note we assume that FWI remains a skillful predictor of area burned (Figure 4), even in a future climate. This assumption is
however highly uncertain due to factors not accounted for in the analysis here, such as possible changes in forest
management (Moreira and Pe'er, 2018; Hudson, 2018), a possible increase / decrease of human-caused forest fires (Balch et
al., 2017) and feedback mechanisms between forest fires and ecology (Balch et al., 2008) .


## 6. Conclusions


In our analysis of the forest fires in Sweden of 2018 we have looked at the risk of fire weather solely on the basis of the Canadian FWI, with the novel approach of using multiple reanalysis datasets and multiple large ensembles with climate models. Using the FWI we have only attributed meteorological aspects of this event, but acknowledge that there are additional aspects important for determining forest fire risk not considered here.


We find that the maximum forest fire risk in July 2018 had return times of ~24 years in Götaland, ~23 years in Svealand and ~24 years in Norrland, with large uncertainty in the reanalysis datasets (90% uncertainty estimate ± ~10 years). Due to the relative short observational record, large uncertainty in the reanalysis datasets and large natural variability of the FWI we cannot infer a robust trend from the reanalysis data alone.


The climate models point to a small increase in risk for such an event at present day compared to pre-industrial conditions for all three regions of about a factor 1.1 (0.9 to 1.4). In a future climate (a $2^0$C warmer climate relative to pre-industrial) the risk for such events to occur may increase more robustly by a factor of ~2 (1.5 to 3) relative to pre-industrial climate according to our model analysis.


The increased fire risk is mostly driven by increased temperature. Though we do find clear changes in precipitation for the warmer climates, we do not see a clear change in prolonged dry periods during summer, which have historically and will likely continue to drive high fire risk events. Our results show the importance of using multiple large ensembles with climate models for attribution studies in order to adequately sample the natural variability and model uncertainties in climate

projections.



**Code availability**

The Python code used to compute the FWI and for the analysis can be obtained by contacting Folmer Krikken (folmer.krikken@knmi.nl)

**Data availability**

EC-Earth data can be obtained by contacting Folmer Krikken (folmer.krikken@knmi.nl), CESM data can be obtained by contacting Flavio Lehner (flehner@ucar.edu) and W@H data can be obtained by contacting Karsten Haustein

(karsten.haustein@ouce.ox.ac.uk).

**Author contribution**

G.J.O. and F.K. conceived and planned the analysis. F.K. provided the EC-Earth data. F.L provided the CESM data. K.H. provided the W@H data. F.K. performed the main analysis and wrote the main manuscript with support from all authors at

all stages.

**Competing interests**

The authors declare that they have no conflict of interests

**Acknowledgements**

F.K. and I.D. contribution was supported by the Belmont Forum Project PREREAL (grant # 292-2015-11-30-13-43-09 to I.D.). I.D. contribution was further supported by the Canadian National Research Council Canada through Discovery Grant (grant # DDG-2015-00026 to I.D.), Swedish Research Council FORMAS (grant # 239-2014-1866 to I.D.), The Swedish Institute funded networks CLIMECO and BalticFire (#10066-2017-13 and #24474/2018 to I.D. The study was conducted

within the framework of the NordicProxy network, which is supported by the Nordic Forest Research (SNS), and consortium GDRI Cold Forests. G.J.O was supported by the ERA4CS project SERV_FORFIRE. F.L. is supported by NSF AGS-0856145, Amendment 87, by the Bureau of Reclamation under Cooperative Agreement R16AC00039, and the Regional and Global Model Analysis (RGMA) component of the Earth and Environmental System Modeling Program of the U.S. Department of Energy's Office of Biological & Environmental Research (BER) Cooperative Agreement DE-FC02-

97ER62402. We thank Guilherme S.J. Pinto for providing the Swedish forest fire data from the MSB.



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





**Table 1: Overview of the climate models and the years used to represent the different climate states.**

| Model | members | Past climate (PI) | Current climate (1ºC) | Future climate ($2^0$C) | Resolution |
|---|---|---|---|---|---|
| **EC-Earth** | 16 | 1900-1950<br>800 yrs total | 1979-2019<br>640 yrs total | 2029-2059<br>480 yrs | 1.1° |
| **CESM1** | 40 | 1920-1950<br> 1200 yrs total | 1987-2027<br>1600 yrs total | 2028-2058<br>1200 yrs | 1° |
| **W@H** | 100 | Natural forcing<br>1986-2015<br>3000 yrs total | Actual forcing<br>1986-2015<br>3000 yrs total | Not available | 0.25° |


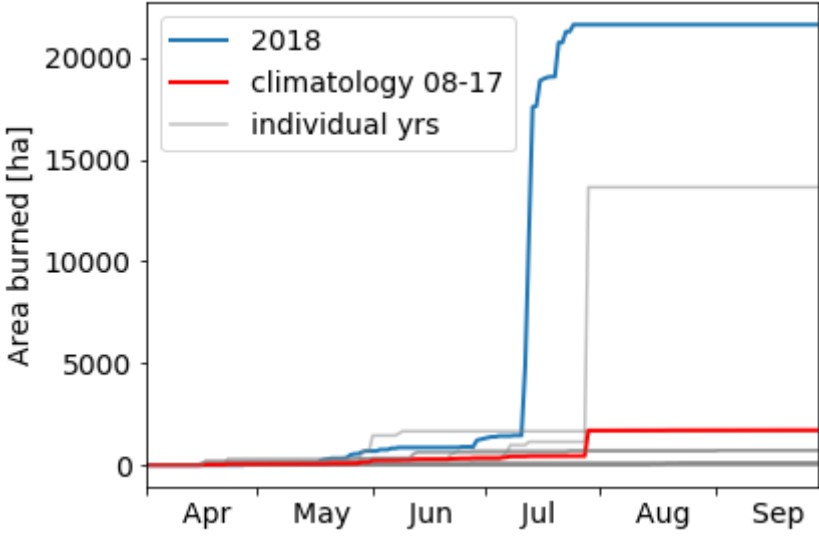


**Figure 1: Area burned in Sweden. Cumulative values for 2018 and climatology over 2008-2017 and its individual years (source: EFFIS).**



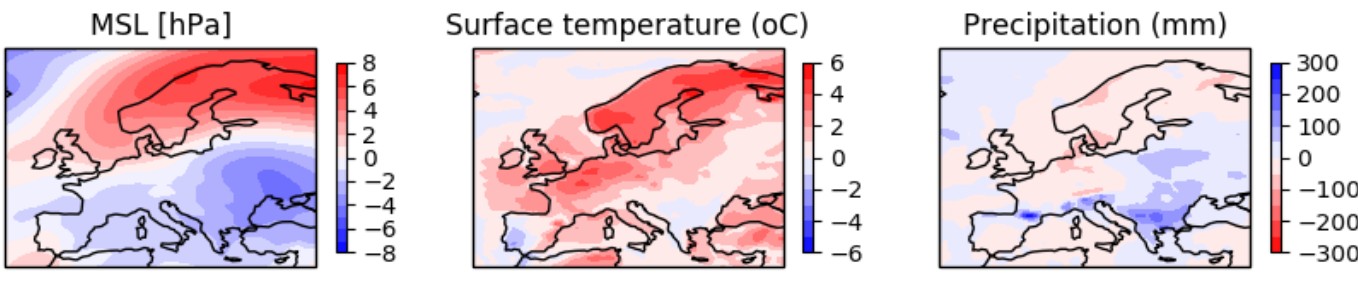

**Figure 2: ERA-Interim July average anomalies of a) mean sea level pressure (MSL), b) surface temperature and c) precipitation. Anomalies are constructed relative to 1981-2010 climatology**





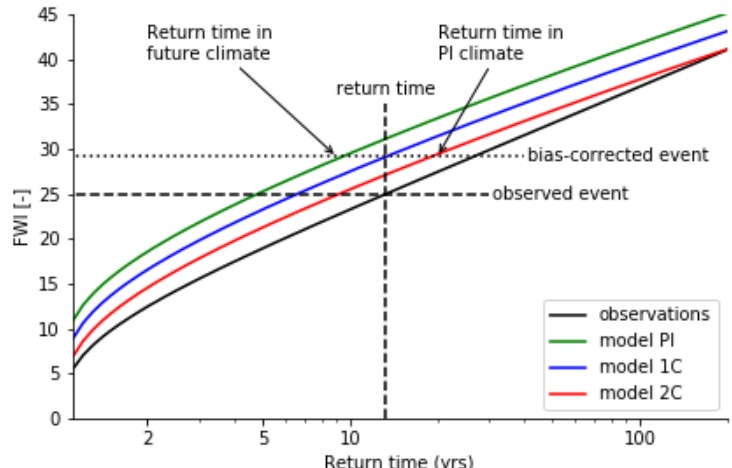

**Figure 3: Schematic of the bias correction method.**

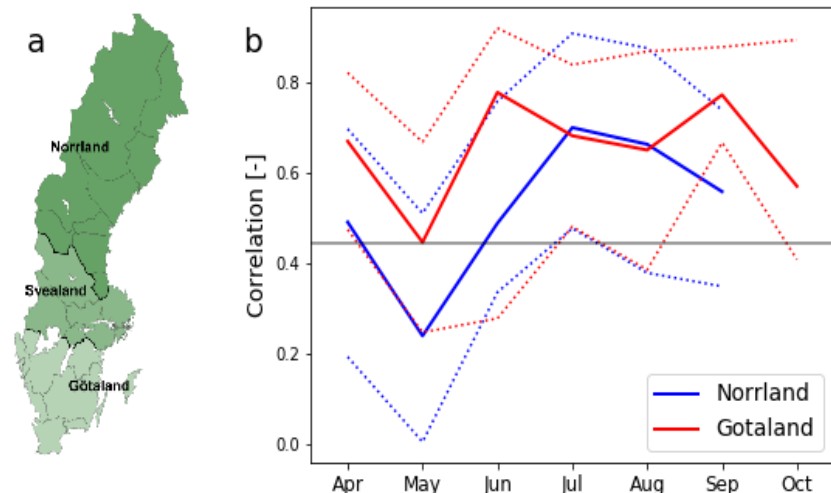

**Figure 4: a) Map of Sweden with the three regions used in this study and b) correlation of FWI (ERA-Interim, monthly maximum value with a 7-day running mean applied) with observed area burned for Norrland and Gotaland from 1998 to 2017. The dotted lines represent the 5-95% bootstrapped confidence intervals and the gray line indicates the significance threshold of 5%. The observed area burned is from Swedish governmental data (MSB, 2016).**
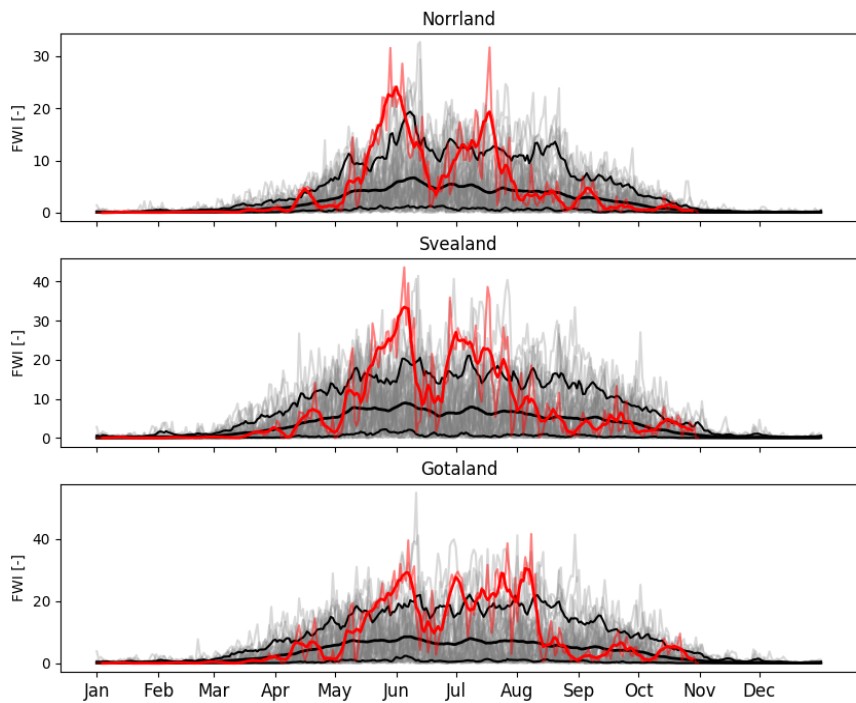

**Figure 5: Area averaged FWI for the three regions (defined in Figure 3). The (thick) red line shows the (7-day running mean) FWI of 2018. The black lines represent the 5, 50 and 95% quantiles of the 1979-2017 climatology and the opaque gray lines the individual years, all based on ERA-Interim extended with ECMWF forecast analysis.**

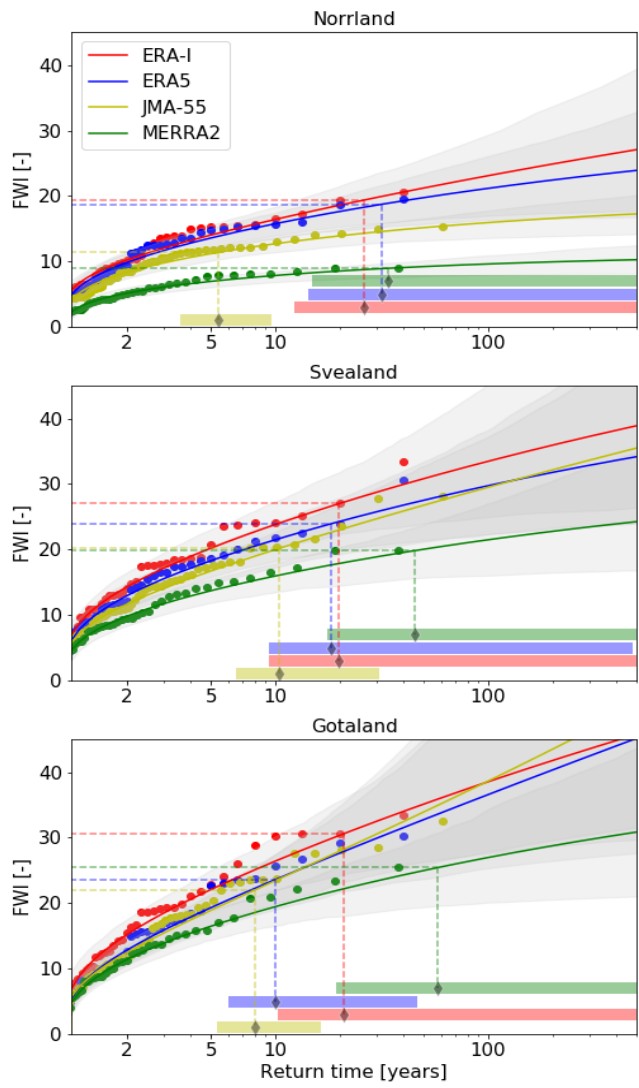

**Figure 6: Return times of July–August maximum FWI values, for all four reanalysis datasets and the three regions. The dots represent the actual FWI maximum values and the lines the GEV model fit with a 5 to 95% uncertainty band in gray. The dashed horizontal lines represents the 2018 event, whilst the vertical line represents the associated return time with the horizontal bars giving the 5% to 95% uncertainty estimate (estimated with a non-parametric bootstrap).**





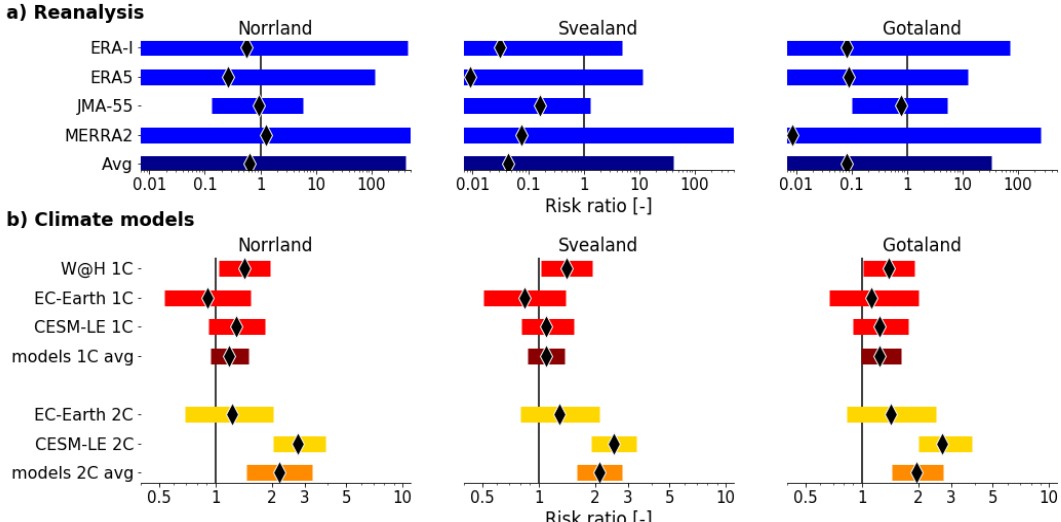

**Figure 7: Risk ratios for maximum July-August FWI values as high as observed in 2018 for the different regions for a) reanalysis and b) climate models. All risk ratios are relative to PI climate. Note the different scales on the x-axis between (a) and (b).**


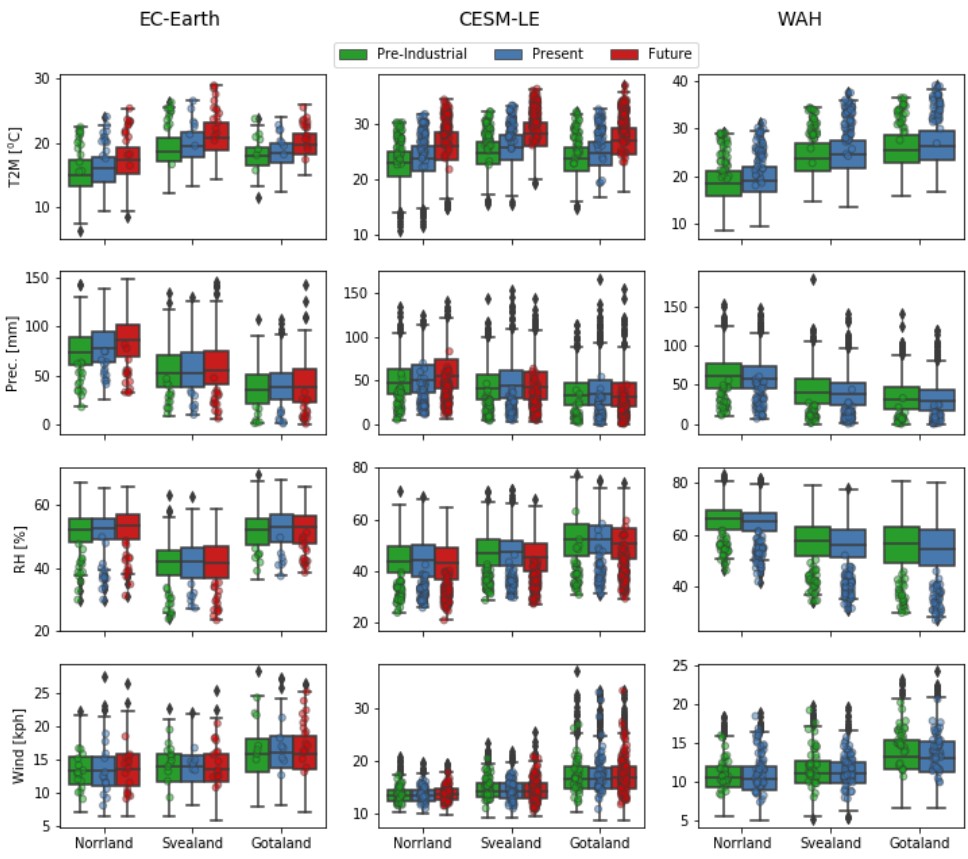

**Figure 8:** Meteorological values associated with the yearly maximum FWI in July and August, with a 7-day rolling average applied, for all three climate states, all three regions. Precipitation is calculated as 30 day cumulative value prior to the yearly maximum FWI. The boxplot shows the quartiles of the distribution, the whiskers the rest of the distribution and the dots are outliers. The round circles indicate all values in the distribution associated with FWI higher than the observed 2018 event.