# Peer review of "Attribution of the role of climate change in the forest fires in Sweden 2018"

_Natural Hazards and Earth System Sciences, 2019_

## Referee Comment (RC1) · Anonymous Referee #1 · 16 Sep 2019

Review of Manuscript submitted to NHESS

Dear Authors, first thank you for a very well-written manuscript about the role of climate change in the forest fires in Sweden 2018. I found the manuscript very interesting; it summarizes quite well the implications of your results and your conclusions are well-defended by the analysis you present. This paper is entirely suitable for publication in NHESS, I have only a few minor suggestions and corrections.

Line 35. I would suggest to include a reference (e.g. Drobyshev et. al. 2012) after "Though forest fires are common in Sweden"

Lines 41-54. There is a recent publication that could be acknowledged: Williams, A. P., Abatzoglou, J. T., Gershunov, A., Guzman‐Morales, J., Bishop, D. A., Balch, J.

[Figure]

K., & Lettenmaier, D. P. (2019). Observed impacts of anthropogenic climate change on wildfire in California. Earth's Future.

Line 107. Please change "it provides" to "they provide"

Line 108. Please change "dataset" to "datasets"

Line 199. Please can you better explain why JRA-55 performs differently?
* * *

---

## Referee Comment (RC2) · Anonymous Referee #2 · 14 Dec 2019

This paper deals with the analysis of model outputs and reanalyses using the FWI as an estimator of fire risk in Sweden.

The approach uses the standard technologies available today – climate scenarios, bias correction and of the like – to perform attribution studies and estimates of fire risk in coming decades.

Overall, however, the main result is that "In a future climate (a 20 C warmer climate relative to pre-industrial) the risk for such events to occur may increase more robustly by a factor of ∼2 (1.5 to 3) relative to pre-industrial climate", which is a result rather similar to many others currently available. In addition, the authors conclude that "The increased fire risk is mostly driven by increased temperature", something couls have been expected also without refined analyses.

[Figure]

In conclusion, I am not convinced this paper is a significant addition to our understanding of fire risk assessment in future climates. If the authors intend to re-submit a new paper, I urge them to develop a critical analysis of the role of the various components (e.g., bias correction) and use different fire indicators.

I think this paper is not acceptable in its current form and I suggest major revision along the lines indicated above.

---

## Author Comment (AC2) · 3 Feb 2020

2-1

This paper deals with the analysis of model outputs and reanalyses using the FWI as an estimator of fire risk in Sweden.

The approach uses the standard technologies available today – climate scenarios, bias correction and of the like – to perform attribution studies and estimates of fire risk in coming decades.

[Figure]

We thank the reviewer for the feedback. We address the individual comments below. A pdf version of this response is attached.

2-2

Overall, however, the main result is that "In a future climate (a 20 C warmer climate relative to pre-industrial) the risk for such events to occur may increase more robustly by a factor ofâĹij2 (1.5 to 3) relative to pre-industrial climate", which is a result rather similar to many others currently available. We agree that, as stated in the manuscript [lines 274-282], our research is in line with other research with the exception of Yang et al (2015) for northern Sweden. Note however that we are the first quantitative attribution study of an extreme forest fire event in this region. Such attribution studies have proven useful in loss and damage policy (James et al. 2019).

The results is of particular relevance as high-latitude European forests are considered to be more sensitive to past and future climate variability (Drobyshev et al 2014), yet quantitative assessments of potential future changes in their disturbance regimes have been missing. The current study fills in this knowledge gap.

2-3

In addition, the authors conclude that "The increased fire risk is mostly driven by increased temperature", something couls have been expected also without refined analyses.

We disagree with this statement as the existence of an expectation is not to be regarded as existence of empirically derived evidence. In fact, many earlier fire studies put the main emphasis on dynamics of precipitation, not temperature (e.g. Lafon & Quiring 2012). An increase in temperature under future climate is indeed expected and it has the potential to yield an increased fire risk. However, just as important in fire weather risk are possible changes in precipitation, specifically changes in consecutive dry days. We mention this aspect in lines 247-255. An a priori assessment of how the

consecutive dry days will change in future climate is not possible as it requires multiple large ensembles with climate models to be assessed robustly. For Sweden, we find no clear changes in consecutive dry days (line 252-254 and figure 8). This allowed us to conclude that temperature is the main driver of the increased fire risk. We believe our analyses contribute with refining our knowledge of future fire risks in Northern Europe.

2-4

In conclusion, I am not convinced this paper is a significant addition to our understanding of fire risk assessment in future climates. If the authors intend to re-submit a new paper, I urge them to develop a critical analysis of the role of the various components (e.g., bias correction) and use different fire indicators.

We agree that a critical analysis of different bias correction methods or the use of multiple fire indicators would be an interesting follow-on study. However, we are under the impression that the manuscript in its current form provides more than enough valuable information for publication, including a number of novel and important methodological considerations. To the best of our knowledge, we are the first to use multiple reanalysis datasets in such an analysis, in which we demonstrated the large differences in FWI and associated return times. This is an important finding with ramifications for other similar research that only uses a single reanalysis product. To the best of our knowledge, we are the first to use multiple large ensemble climate models for such an analysis. Events such as the Sweden forest fires in 2018 have been, as demonstrated in this manuscript, rather extreme ($\sim$25 years return time). Hence, in order to acquire robust statistics on these events, and to extract a possible climate change signal, we need a large ensemble. This approach is particularly relevant in relation to precipitation dynamics and the resulting precipitation changes as its projections are highly model dependent. Thus for an adequate sampling of the model uncertainty we need multiple climate models. Previous research is often limited by both or one of these factors.

I think this paper is not acceptable in its current form and I suggest major revision along the lines indicated above.

We regret this assessment and humbly disagree. We hold that we are providing novel contributions to both the assessment of fire risk over Sweden as well as novel way to quantify methodological uncertainties in such assessments. We agree that exploring different bias correction schemes would be another of many additional analyses that could be conducted in a follow-up study. To address this potential, we now include a short discussion of this in the Discussion section of our paper [lines 285-289 in file with track changes, lines 284-288 in file without track changes].

We would like to sincerely thank both reviewers for many stimulating comments, which helped improve the quality of presentation.

References:

Drobyshev,I., Granström,A., Linderholm,H.W., Hellberg,E., Bergeron,Y. & Niklasson,M. 2014. Multi-century reconstruction of fire activity in Northern European boreal forest suggests differences in regional fire regimes and their sensitivity to climate. Journal of Ecology 102: 738-748.

James, Rachel & Jones, Richard & Boyd, Emily & Young, Hannah & Otto, Friederike & Huggel, Christian & Fuglestvedt, Jan. (2019). Attribution: how is it relevant for loss and damage policy and practice?. https://link.springer.com/chapter/10.1007/978-3-319-72026-5_5 Lafon CW and Quiring SM 2012. Relationships of Fire and Precipitation Regimes in Temperate Forests of the Eastern United States. Earth Interactions 16: 1-15.

Sillmann, J., Kharin, V. V., Zwiers, F. W., Zhang, X., and Bronaugh, D. ( 2013), Climate extremes indices in the CMIP5 multimodel ensemble: Part 2. Future climate projections, J. Geophys. Res. Atmos., 118, 2473– 2493, doi:10.1002/jgrd.50188.

Yang, W., Gardelin, M., Olsson, J. and Bosshard, T.: Multi-variable bias correction: application of forest fire risk in present and future climate in Sweden, Nat. Hazards Earth Syst. Sci., 15(9), 2037–2057, doi:https://doi.org/10.5194/nhess-15-2037-2015, 2015

Please also note the supplement to this comment:
https://www.nat-hazards-earth-syst-sci-discuss.net/nhess-2019-206/nhess-2019-206-AC2-supplement.pdf

---

## Author Response (AR2)

Review of Manuscript submitted to NHESS

Dear Authors, first thank you for a very well-written manuscript about the role of climate change in the forest fires in Sweden 2018. I found the manuscript very interesting; it summarizes quite well the implications of your results and your conclusions are well-defended by the analysis you present. This paper is entirely suitable for publication in NHESS, I have only a few minor suggestions and corrections.

We thank the reviewer for the positive feedback. We address the individual comments below. The mentioned line numbers relate to the updated manuscript with track changes.

1-1
Line 35. I would suggest to include a reference (e.g. Drobyshev et. al. 2012) after "Though forest fires are common in Sweden"

We have added the reference as suggested

1-2
Lines 41-54. There is a recent publication that could be acknowledged: Williams, A.P., Abatzoglou, J. T., Gershunov, A., Guzmanã˘AˇRMorales, J., Bishop, D. A., Balch, J.C1K., & Lettenmaier, D. P. (2019). Observed impacts of anthropogenic climate change onwildfire in California. Earth's Future.

We have added the reference as by including the line: '*Williams et al. (2019) found a strong influence between recent increase in forest fires in California and the positive trend in vapor pressure deficit cause by anthropogenic climate change.*'

1-3
Line 107. Please change "it provides" to "they provide"

We have changed the sentence as suggested

1-4
Line 108. Please change "dataset" to "datasets"

Corrected as suggested

1-5
Line 199. Please can you better explain why JRA-55 performs differently?

JMA-55 performs differently because it has a longer time record since it starts in 1955 where the other datasets start in 1979 or 1980. The uncertainty estimates of the return times largely depends on the length of record. All other reanalysis products cover a shorter time period, thus their uncertainties are larger. Importantly, the JRA-55 uncertainty range is contained within the uncertainty range of the other reanalysis products, so the estimates from JRA-55 and the other products are consistent.
To improve the wording we have changed the sentence from '*… (except JRA-55 that includes more data) …* ' to '*…(except JRA-55 where the variability is based on a 25 year longer timeseries)…*'

*We would like to sincerely thank both reviewers for many stimulating comments, which helped improve the quality of presentation.*

Anonymous Referee #2

2-1
This paper deals with the analysis of model outputs and reanalyses using the FWI as an estimator of fire risk in Sweden.

The approach uses the standard technologies available today – climate scenarios, bias correction and of the like – to perform attribution studies and estimates of fire risk in coming decades.

We thank the reviewer for the feedback. We address the individual comments below. The mentioned line numbers relate to the updated manuscript with track changes.

2-2
Overall, however, the main result is that "In a future climate (a 20 C warmer climate relative to pre-industrial) the risk for such events to occur may increase more robustly by a factor of~2 (1.5 to 3) relative to pre-industrial climate", which is a result rather similar to many others currently available.

We agree that, as stated in the manuscript [lines 286-299], our research is in line with other research with the exception of Yang et al (2015) for northern Sweden. Note however that we are the first quantitative attribution study of an extreme forest fire event in this region. Such attribution studies have proven useful in loss and damage policy (James et al. 2019).

The results is of particular relevance as high-latitude European forests are considered to be more sensitive to past and future climate variability (Drobyshev et al 2014), yet quantitative assessments of potential future changes in their disturbance regimes have been missing. The current study fills in this knowledge gap.

2-3
In addition, the authors conclude that "The increased fire risk is mostly driven by increased temperature", something couls have been expected also without refined analyses.

We disagree with this statement as the existence of an expectation is not to be regarded as existence of empirically derived evidence. In fact, many earlier fire studies put the main emphasis on dynamics of precipitation, not temperature (e.g. Lafon & Quiring 2012). An increase in temperature under future climate is indeed expected and it has the potential to yield an increased fire risk. However, just as important in fire weather risk are possible changes in precipitation, specifically changes in consecutive dry days. We mention this aspect in lines 260-264. An a priori assessment of how the consecutive dry days will change in future climate is not possible as it requires multiple large ensembles with climate models to be assessed robustly. For Sweden, we find no clear changes in consecutive dry days (line 265-267 and figure 8). This allowed us to conclude that temperature is the main driver of the increased fire risk. We believe our analyses contribute with refining our knowledge of future fire risks in Northern Europe.

2-4
In conclusion, I am not convinced this paper is a significant addition to our understanding of fire risk assessment in future climates. If the authors intend to re-submit a new paper, I urge them to develop a critical analysis of the role of the various components (e.g., bias correction) and use different fire indicators.

We agree that a critical analysis of different bias correction methods or the use of multiple fire indicators would be an interesting follow-on study. However, we are under

the impression that the manuscript in its current form provides more than enough valuable information for publication, including a number of novel and important methodological considerations.

To the best of our knowledge, we are the first to use multiple reanalysis datasets in such an analysis, in which we demonstrated the large differences in FWI and associated return times. This is an important finding with ramifications for other similar research that only uses a single reanalysis product.

To the best of our knowledge, we are the first to use multiple large ensemble climate models for such an analysis. Events such as the Sweden forest fires in 2018 have been, as demonstrated in this manuscript, rather extreme (~25 years return time). Hence, in order to acquire robust statistics on these events, and to extract a possible climate change signal, we need a large ensemble. This approach is particularly relevant in relation to precipitation dynamics and the resulting precipitation changes as its projections are highly model dependent. Thus for an adequate sampling of the model uncertainty we need multiple climate models. Previous research is often limited by both or one of these factors.

2-5

I think this paper is not acceptable in its current form and I suggest major revision along the lines indicated above.

We regret this assessment and humbly disagree. We hold that we are providing novel contributions to both the assessment of fire risk over Sweden as well as novel way to quantify methodological uncertainties in such assessments. We agree that exploring different bias correction schemes would be another of many additional analyses that could be conducted in a follow-up study. To address this potential, we now include a short discussion of this in the Discussion section of our paper [lines 295-299].

We would like to sincerely thank both reviewers for many stimulating comments, which helped improve the quality of presentation.

Point by point response to editor comments.

We thank the editor for the constructive feedback on the manuscript. Please find below a point by point response to the feedback.

Point 1
- to update the state of the art that you show in the Introduction (see, for instance, the papers
published in the last five years in Nature Communications, Nature Climate Change, Climate
Change,..., about forest fires and climate), in order to compare your results (other papers show a
decreasing trend in forest fires- see for instance Turco M, Bedia J, Di Liberto F, Fiorucci P, von
Hardenberg J, Koutsias N, et al. (2016) Decreasing Fires in Mediterranean Europe. PLoS ONE
11(3): e0150663. doi:10.1371/journal.pone.0150663);

We have updated our literature section with several papers, including Turco et al 2016. See the 2nd paragraph of the introduction for the added literature.

Point 2
- to avoid conclusions like the fact that the forest fires produced in Sweden in 2018 have been consequence of climate change, when you have obtained a decreasing trend until 2017

Please note that we do not state that the forest fires in Sweden 2018 have been consequence of climate change. Neither in the abstract nor the conclusion. However, we can understand that the last line of the abstract can be interpreted as such, hence we have removed this sentence ('In summary, … the future'). Also, we have now added '(non-significant)' to the line 23 in order to show that the results of the increased probability (current vs PI climate) are not significant.

Point 3
- … to include explanations about this observed decreasing trend

We have now expanded on this in section 4.1, by including the sentence: 'A trend analysis of the FWI input variables during highFor FWI events reveals a negative trend in wind speed, a positive trend in local noon surface temperature, and a positive trend in 30-day cumulative precipitation prior to high FWI events. The net effect on the FWI is thus a small negative trend.

Point 4
- … to explain better how do you obtain the factors of increased risk and what is the meaning of the expression "increased risk" (increased probability?, increased extension?, increased number of forest fires?)

We agree it is not entirely clear what is meant by 'risk'. In order to clarify we have rephrased risk to 'probability' throughout the manuscript. This is explained in detail in the last part of section 2.2. In some instances we still use the phrase 'fire weather risk'. Here we mean in an increase in the FWI (which is explained in detail in section 2.1).

The line numbers refer to the manuscript with track changes.

- Changed the corresponding author to geert.jan.van.oldenborgh@knmi.nl.
- Replaced 'risk' with 'probability' where a quantified risk is meant throughout the manuscript.
- Replaced JMA with JRA, correction of the abbreviation
line 29-30: Removed the last line from the abstract.
Line 50-59: Added relevant references / literature.
Line 207-208: Replaced 'that includes more data' with 'where the variability is based on a 25 year longer timeseries'
Line 220-229: Improved description of why there is a slight negative trend in the FWI of the reanalysis products
Line 294-299: Added more options for where future work could focus on.

3000 yrs total | Not available | 0.25° |

490

[Figure]

**Figure 1: Area burned in Sweden. Cumulative values for 2018 and climatology over 2008-2017 and its individual years (source: EFFIS).**

495

[Figure]

**Figure 2: ERA-Interim July average anomalies of a) mean sea level pressure (MSL), b) surface temperature and c) precipitation. Anomalies are constructed relative to 1981-2010 climatology**

500

[Figure]

**Figure 3: Schematic of the bias correction method.**

[Figure]

**Figure 4: a) Map of Sweden with the three regions used in this study and b) correlation of FWI (ERA-Interim, monthly maximum value with a 7-day running mean applied) with observed area burned for Norrland and Gotaland from 1998 to 2017. The dotted lines represent the 5-95% bootstrapped confidence intervals and the gray line indicates the significance threshold of 5%. The observed area burned is from Swedish governmental data (MSB, 2016).**

[Figure]

**Figure 5: Area averaged FWI for the three regions (defined in Figure 3). The (thick) red line shows the (7-day running mean) FWI of 2018. The black lines represent the 5, 50 and 95% quantiles of the 1979-2017 climatology and the opaque gray lines the individual years, all based on ERA-Interim extended with ECMWF forecast analysis.**

[Figure]

**Figure 6: Return times of July–August maximum FWI values, for all four reanalysis datasets and the three regions. The dots represent the actual FWI maximum values and the lines the GEV model fit with a 5 to 95% uncertainty band in gray. The dashed horizontal lines represents the 2018 event, whilst the vertical line represents the associated return time with the horizontal bars giving the 5% to 95% uncertainty estimate (estimated with a non-parametric bootstrap).**

[Figure]

520

**Figure 7:** Probability ratios for maximum July-August FWI values as high as observed in 2018 for the different regions for a) reanalysis and b) climate models. All probability ratios are relative to PI climate. Note the different scales on the x-axis between (a) and (b).

[Figure]

525

**Figure 8: Meteorological values associated with the yearly maximum FWI in July and August, with a 7-day rolling average applied, for all three climate states, all three regions. Precipitation is calculated as 30 day cumulative value prior to the yearly maximum FWI. The boxplot shows the quartiles of the distribution, the whiskers the rest of the distribution and the dots are outliers. The round circles indicate all values in the distribution associated with FWI higher than the observed 2018 event.**

530

---

## Author Response (AR3)

<h1 style="text-align:center">Point to point response on reviewer 3</h1>

Dear Mr. Krikken and coauthors

Thank you very much for your interesting paper "Attribution of the role of climate change in the forest fires in Sweden 2018". I think that your paper can be accepted after doing some minor changes.

*We thank the reviewer for investing the time for this in-depth review. Please find the point-by-point response to the comments below. Line numbers refer to the manuscript with track changes.*

*Abstract*

*Please, modify the abstract. It provides contradictory information. In the first paragraph you say "we find a negative trend of the FWI for Southern Sweden over the 1979 to 2017 time period" but the second paragraph ends with "We however do not find a clear change of prolonged dry periods in summer months that could explain the increased fire weather risk.". Perhaps this second conclusion refers to climate projections but, in this case, you should clarify it.*

Response: Thank you for pointing this out. We have clarified this point in the text as indeed the second paragraph point to the results of the climate projections. We have now added "*in the climate models*" at the end of the abstract and clarified the text more.

*Figure 1 and so on. It is "burned area" not "area burned". Please, modify.*

Response: We have no changed all 'area burned' to 'burned area'

*Figure 1. What do you mean with "climatology over 2008-2017"? Do you refer to average annual value of burned area? What do you mean with "its individual years" Please, clarify the meaning of both terms in the main text.*

Response: The climatology refers to average cumulative values over the year, based on the data from 2008-2017. Hence, the climatology is also cumulative. The individual years all shows the cumulative values of burned area for each individual year. We have clarified this in the new figure caption on line numbers 513-514.

*"Figure 1: Burned area in Sweden. Cumulative values for 2018, average cumulative value (climatology) over 2008-2017 and the cumulative values for each individual year over the same time period (source: EFFIS)."*

*Figure 4. You only work in the paper with two Sweden regions, then you cannot say "Map of Sweden with the three regions used in this study". On the other hand, what is the criteria of this regionalization? Are they climate regions? Please, modify the text as convenient.*

Response: Thank you for pointing this out. We believe the 3 regions in figure 4a together with only 2 lines in Figure 4b may have caused some confusion. Our analysis is indeed performed over 3 regions, as

stated in lines 149-151. For the analysis in Figure 4b we excluded Svealand (middle Sweden) because there were not enough fires / burned area to calculate robust statistics. This is mentioned in lines 184 to 185. However, to prevent further confusion on this we also added this information in the figure caption. Note that we do use Northern and Southern Sweden in the abstract to make in easier to understand for readers that do not read the complete manuscript.

The reason for splitting up Sweden is indeed not explicitly mentioned. We refer the reader to Drobyshev et al., 2012, as stated in lines 158-159.  However, we agree that the manuscript would benefit in a short description. We have added the following explanation the manuscript: *"Division of Sweden into Southern and Northern parts was justified by the analysis of observational fire statistics (Drobyshev et al., 2012) and dendrochronological reconstruction (Drobyshev et al., 2014) that revealed limited synchrony in the annual fire activity between these two regions."*

*Line 49. You say: "found a strong influence between the recent increase in forest fires in California and the positive trend in vapor pressure deficit caused by anthropogenic climate change." However, the temperature increase implies an increase of saturate vapor pressure and evaporation. Consequently, there is an increase of evapotranspiration and you can write "and the positive trend in evapotranspiration caused…."*

Response: Thank you for pointing this out. It has now been corrected.

*Line 59. Please, add a short explanation to justify the reason of this difference between Southern Sweden and Northern Sweden. Afterwards, in line 145 you say "Since Southern Sweden has a different fire climate than Northern Sweden" but for the reader that does not know Sweden, a short explanation is needed.*

Response: This has been addressed in the other comment about division between Northern and Southern sweden.

*Line 91. Delete the second bracket in "(GISTEMP, (Hansen et al., 2010)".*

Response: This has been corrected.

*Line 119. Delete the second bracket in "(CESM1, (Kay et al., 2014)".*

Response: This has been corrected.

*Line 121. Replace RCP85 by RCP8.5*

Response: This has been corrected.

*Line 132. You say: "For the W@H simulations the GMST increase between the 'natural forcing' simulations and the 'actual forcing' simulations is 0.65ºC, which is very close to the observed warming."*

*But the last AR5 IPCC report showed that the warning was 0.85ºC and nowadays it is near 1ºC. The best will be to clarify in this sentence the period to which you refer when you say "observed warning".*

Response: The AR5 IPCC indeed reports a higher warming. This is because our 'current climate' is based on the 1979-2018 average, whereas AR5 IPCC bases it on the linear trend up to 2010. We chose this time period so we can best compare the climate models to the reanalysis (that cover 1979-2018). This is mentioned in lines 136-137. We have added 'up to 1979-2019' after line 143-144 to further clarify this.

*Line 142. In this case references are written without comma "(Ho et al. 2011; Ehret et al. 2012)", but in the major part of the paper you add a comma after the dot. Please, review all the references cited in the text in order to homogenize them and adding the comma if necessary.*

Response: Thank you for pointing this out. We now homogenized all references to using a comma.

*Lines 272-279. Please, delete or modify completely this paragraph. The non-linear model behavior observed or modeled is consequence of the non-linearity of atmospheric process and cannot be only related to the non-linearity of radiative processes. There exists a lot of discussion about this fact related with the dimming phase, mainly if you refer to the decade of 50's when the anthropogenic climate change effect was still minor. There are a lot of factors related with forest fires production and non-linearity because they are also related with the potential combustible fuel. Vegetation has its own growth biorhythms that can depend of precipitation or temperature in other seasons of the year, and the climate change impact can be different for the different seasons of the year. I recommend you reading the papers from Turco et al., 2014 and Turco et al., 2018. The first one provides information about the different factors that can act in the forest fire risk and burned area. The second one can help you in the discussion about future scenarios*

*Turco, M., M.C. Llasat, J. von Hardenberg, A. Provenzale, 2014. Climate change impacts on wildfires in a Mediterranean environment. Climatic Change 125:369–380. DOI 10.1007/s10584-014-1183-3*
*Turco, M., J. J.Rosa-Cánovas, J. Bedia, S. Jerez, J. P. Montávez, M. C. Llasat and A. Provenzale, 2018. Exacerbated fires in Mediterranean Europe due to anthropogenic warming projected with nonstationary climate-fire models. NATURE COMMUNICATIONS | (2018) 9:3821 | DOI: 10.1038/s41467-018-06358-z | www.nature.com/*

Response: Thank you for clarifying this. We have decided to remove this part of the manuscript.

*You obtain strong differences using the different models. Then it would be necessary to add a comparison of the FWI outputs from the models with forest fire observed data (burned area or number of forest fires) for the longest available period. I would recommend you in order to improve the paper and having a major impact, to add a figure with this information and discuss it in the results or discussion sections.*

Response: Thank you for pointing this out. It is indeed very important to validate the climate models to observations. We did perform a validation on the GEV fit by comparing the shape and scale parameter of

the GEV fit to the ones from ERA5. We found that these lie within the uncertainty estimates of the shape and scale parameter of ERA5. This was indeed not mentioned in the manuscript. We have now added this to section 2.4 (line nr 176-177): *"The models are further validated by comparing the scale and shape parameter of the GEV-fit to the ERA5 GEV-fit. All model parameters lie withing the uncertainty estimate of the ERA parameters."*

Comparing the FWI output from the model to observed fire data as suggested by the reviewer will unfortunately not provide the answers to this question. The natural variability in the climate models is not in phase with the observed climate / fire variability, hence a comparison as done in Figure 4b will not be valid beyond the trends, which are already analysed in Fig. 7.

We went through the entire manuscript again for a final check and made some small changes to correct small typos, further clarify results by including confidence intervals and to improve readability. Note that these changes do not influence the conclusions or results in any way. The changes are listed below. The line number correspond to the manuscript with track changes.

Line nr 13: Changed the correspondece person to Geert Jan van Oldenborgh

Line nr 19: Added cofidence intervals around the return time '*(90% CI > 10yr)*'

Line nr 20-21: Change to improve readibility, changed '*.. time period, yielding a decreasing probability of such an event solely based on reanalysis data.*' to '*.. time period in the reanalyses, yielding a non-significant reduced probability of such an event.*'.

Line nr 21: Removed '*given*'.

Line nr 22-23: Added text to highlight the uncertainty of the results, added '*.. give a large confidence interval around the number that easily includes no change, so..*'

Line nr 24: Removed '*on the other hand*'

Line nr 24: Added confidence interval '*(0.9 to 1.4)*'

Line nr 25: Added confidence interval '*(1.5 to 3)*'

Line nr 33-35: Added a short summary to the abstract '*In summary, we find a (non-significant) reduced probability of such events based on reanalyses. but a small (non-significant) increased probability due to global warming up to now and a more robust (significant) increase in the risk for such events in the future based on the climate models.*'

Line nr 98-99: Small correction on the data used, changed '*4th order polynomial*' to '*4yr running mean*'.

Line nr 105: Small correction. Changed '*.. dependence of the FWI the same ..*' to '*.. the dependence of the smoothed GMST the same ..*'.

Line nr 106: Added '*van der*' to the referens.

Linde nr 107: Changed '*precipitation*' to '*extreme precipitation*'

Line nr 108: Added '*to keep the FWI positive-definite*'.

Line nr 113 and 114: Changed '*will be*' to '*is*'.

Line nr 239: Textual change, changed '*The net effect on the FWI is thus a ..*' to '*The resulting net effect on the FWI is a ..*'

Line nr 245: Textual change, changed '*data*' to '*output*'

Line nr 253: Added a confidence interval, added '*factor 1.1 with a 90% CI of 0.9 to 1.4*'

Line nr 255: Added  a confidence interval, added '*with a 90% CI of 1.5 to 3*'

Line nr 258: Added a confidence interval, added '*(0.9 to 1.4)*'

Line nr 259: Added a confidence interval, added '*(1.5 to 3)*'

Line nr 299: Textual change, changed '*Another*' to '*a*'

Line nr 309: Added a citation ('*Hauser et al., 2017*')

Line nr 316: Textual change, added '*that*'

Line nr 326-327: Added examples to additional aspects for determining forest fire risk that were not considered. Added '*such as ignition sources, forest management and ecology*'

Line nr 330: Textual change, changed '*90% uncertainty estimate ~10 years*' to '*the 90% confidence interval starts at ~10 years*'

Line nr 334: Added '*an insignificant*'